



**Estimation of rainfall erosivity based on WRF-derived raindrop size distributions**
Qiang Dai[1, 2], Jingxuan Zhu[1], Shuliang Zhang[1], Shaonan Zhu[3], Dawei Han[2] and Guonian Lv[1]
[1]Key Laboratory of VGE of Ministry of Education, Nanjing Normal University, Nanjing, China.
[2]Department of Civil Engineering, University of Bristol, Bristol, UK.
[3]College of Geographical and Biological Information, Nanjing University of Posts and
Telecommunications, Nanjing, China
Corresponding author: Qiang Dai (qd_gis@163.com)
**Key Points:**
- WRF-derived rainfall kinetic energy offers a novel way to estimate large-scale soil
erosion.
- Annual rainfall and erosivity are not always positively correlated.
- Highest rainfall erosivity of UK occurs in the west coast area during 2013-2017.





**Abstract**

Soil erosion can cause various ecological problems, such as land degradation, soil fertility loss, and river siltation. Rainfall is the primary water-driving force for soil erosion and its potential effect on soil erosion is reflected by rainfall erosivity that relates to the raindrop kinetic energy (KE). As it is difficult to observe large-scale dynamic characteristics of raindrops, all the current rainfall erosivity models use the function based on rainfall amount to represent the raindrops KE. With the development of global atmospheric re-analysis data, numerical weather prediction (NWP) techniques become a promising way to estimate rainfall KE directly at regional and global scales with high spatial and temporal resolutions. This study proposed a novel method for large-scale and long-term rainfall erosivity investigations based on the Weather Research and Forecasting (WRF) model, avoiding errors caused by inappropriate rainfall–energy relationships and large-scale interpolation. We adopted three microphysical parameterizations schemes (Morrison, WDM6, and Thompson aerosol-aware [TAA]) to obtain raindrop size distributions, rainfall KE and rainfall erosivity, with validation by two disdrometers and 304 rain gauges around the United Kingdom. Among the three WRF schemes, TAA had the best performance compared with the disdrometers at a monthly scale. The results revealed that high rainfall erosivity occurred in the west coast area at the whole country scale during 2013-2017. The proposed methodology makes a significant contribution to improving large-scale soil erosion estimation and for better understanding microphysical rainfall–soil interactions to support the rational formulation of soil and water conservation planning.

# 1 Introduction

Soil erosion has a pivotal role in shaping the Earth's physical landscape; however, it can threaten both ecosystems and human societies (Alewell et al., 2015). Accurate quantification of soil loss impact at large spatial scales is therefore important for developing land-use planning and sustainable conservation practices (Bilotta et al., 2012). The soil erosion rate is driven by a combination of factors, which include rainfall, topography, soil characteristics, land cover, and land management applications (Wischmeier and Smith, 1958; Panagos et al., 2015b). Among these, rainfall is a driving force that accounts for a large proportion of soil loss throughout most of world (Panagos et al., 2015b). The erosive force of rainfall with consequent runoff is





represented as erosivity of rainfall, which is a crucial factor for estimating soil loss in large-scale
soil erosion models; for instance, the Universal Soil Loss Equation (USLE (Wischmeier and
Smith, 1978) or RUSLE (Renard et al., 1997)), Limburg Soil Erosion Model (LISEM) (De Roo
et al., 1996), and USLE-M (Kinnell and Risse, 1998).
Rainfall erosivity estimation involves the microphysical properties of rainfall and
rainfall–soil interactions on different time steps (Petan et al., 2010). Impact of rainfall, the main
mechanism driving the splashing of soil particles from the soil mass, which leads to soil erosion
through soil disintegration and mobilization, relies on the kinetic energy (KE) of raindrop
motions (Wischmeier and Smith, 1958; Wang et al., 2014). Robust measurement of raindrop size
and terminal velocity is vital for estimating and predicting rainfall erosivity. Many measurements
can be used to obtain these two parameters, including the stain paper or flour pellet methods
(Marshall and Palmer 1948; Wischmeier and Smith, 1958), high speed cameras (Jones, 1959;
Kinnell, 1981; McIsaac, 1990), and disdrometers (Petan et al., 2010; Angulo-Martinez et al.,
2012). Accurate measurements of raindrop size can be provided in all their methods, and
terminal velocity of raindrops can be further measured by video cameras and disdrometers.
Velocity can also be estimated as the function of raindrop diameter from the empirical
relationship (Beard, 1976; Atlas and Ulbrich, 1977; Uplinger, 1981; Van Dijk et al., 2002).
When using ground observations, rainfall KE can be estimated at a given site.
However, direct measurement of rainfall KE in a large area is difficult because it requires
considerable effort, as well as a dense network of expensive instruments that provide accurate
outputs (Fornis et al., 2005; Mikoš et al., 2006; Meshesha et al., 2016; Dai et al., 2017). Previous
studies have therefore mainly employed more readily accessible records like rainfall intensity,
and attempted to estimate rainfall KE from the empirical relationship of unit KE ($ke$) with
intensity ($ke$–$I$). Since Marshall and Palmer (1948) first observed a two-parameter exponential
relationship between drop size and intensity, several forms of $ke$–$I$ mathematical expressions for
specific locations and climatic conditions have been proposed, including power-law (Park et al.,
1982; Meshesha et al., 2016), linear (Sempere-Torres et al., 1998; Nyssen et al., 2005),
polynomial (Carter et al., 1974), logarithmic (Wischmeier and Smith, 1978; Davison et al., 2005;
Meshesha et al., 2014), and exponential (Kinnell, 1981; Brown and Foster, 1987) relationships.
Among these, the exponential function has been preferentially used currently (Van Dijk et al.,
2002; Fornis et al., 2005; Petan et al., 2010; Sanchez-Moreno et al., 2012; Lim et al., 2015).



Accurate raindrop size distribution (DSD) measured by disdrometers is widely used to derive $ke$–
$I$ relationships (Angulo-Mart ńez et al., 2016; Meshesha et al., 2016). However, such empirically
derived formulas indicate that rainfall $ke$ will increase infinitely with increasing intensity,
whereas studies (Rosewell, 1986; Angulo-Mart ńez et al., 2016; Meshesha et al., 2019) have
found that rainfall $ke$ reaches an top value when intensity is around 70 mm h$^{-1}$ (Hudson, 1963;
Wischmeier and Smith, 1978). More importantly, such a $ke$–$I$ relationship only represents local
climate and precipitation microphysics, and is valid for such regions. There is great uncertainty
associated with rainfall erosivity estimation using this $ke$–$I$ relationship in a large domain
(Angulo-Mart ńez and Barros, 2015), especially due to the poor spatial and temporal
predictability of the $ke$–$I$ relationship. This has motivated researchers to directly calculate KE
based on large-scale DSD measurements.
Ground- and space-based radar can be used to obtain DSD parameters (Atlas et al., 1973;
Doelling et al., 1998). For example, the space-borne Dual-frequency Precipitation Radar (DPR)
radar containing Ku- and Ka-bands in the Global Precipitation Measurement (GPM) satellite
allows researchers to estimate the global three-dimensional spatial distribution of hydrometeors.
Unfortunately, ground dual-polarization radars are available in limited areas (Prigent, 2010) with
large uncertainties (Dai et al., 2019), and the GPM DPR instrument, which measures DSD with
daily or longer temporal resolutions, fail to capture a full storm and meet the requirement for
rainfall kinetic estimation. Mesoscale numerical weather prediction models, for instance, the
WRF model, can simulate microphysical cloud processes and predict the evolution of particle
size distribution through computationally feasible parametrization schemes (Dai et al., 2014;
Brown et al., 2016). DSD on the ground can be derived from the WRF model through
consideration of various physical processes, types of hydrometeor, and free degrees of size
distributions in hydrometeor. As such, a number of recent researches have investigated the
retrieval and uncertainty of DSD parameters by WRF (Gilmore et al., 2004; Ćurić et al., 2009;
Brown et al., 2016; Yang et al., 2019).
The WRF model runs with initial and boundary conditions using global reanalysis
datasets, such as those of the European Centre for Medium-range Weather Forecasts (ECMWF)
and National Centers for Environmental Prediction (NCEP). In other words, WRF-derived DSD
can be obtained for any given area with fine spatial and temporal resolutions rather than
traditional course linear interpolations. We therefore attempted to estimate rainfall erosivity for



the whole United Kingdom (UK) domain using WRF-derived DSD. For comparison, we
calculated interpolated traditional disdrometer-derived rainfall erosivity. To our knowledge, this
work is the first attempt to take advantage of a numerical weather prediction model for
estimating rainfall erosivity anywhere around the world. The current study contributes to the
development of large-scale soil erosion estimation and provides a better comprehension of
microphysical rainfall–soil interactions.
**2 Methodology**
2.1 Disdrometer-based rainfall KE estimation
KE dominates the ability of raindrop to separate soil particles. The KE ($e$, unit: J) of a
raindrop with mass $m$ (g) and terminal velocity $v$ (m s$^{-1}$) is defined by:

$$e = \frac{1}{2} mv^2 \tag{1}$$

Assuming a spherical volume for every raindrop shape, the mass of a drop can be
calculated from the cube of the diameter $D$ (mm). Because instruments (e.g., disdrometers)
generally sample drop size, the mean radius and falling velocity of the corresponding sampling
drop-size class is used to represent $D$ and $v$, expressed as $D_i$ and $v_i$, respectively. In such cases,
the $e_i$ with any drop of a given class is given as:

$$e_i = \frac{1}{12} 10^{-6} \pi \rho v_i^2 D_i^3 \tag{2}$$

where $\rho$ is the water density (g cm$^{-3}$). The sum of the KE of each individual raindrop within a
given rain depth that hits a given area defines the total KE. The unit rainfall KE $ke_t$ in the $t^{\text{th}}$
minute (MJ ha$^{-1}$ mm$^{-1}$) can be calculated as the sum of each drop KE in each size set, as follows:

$$ke_t = \frac{e_{sum}}{AP_t} = \frac{1}{AP_t} \sum_{i=1}^{ni} N_i e_i \tag{3}$$

where $A$ represents the sample area of the sensor, $P_t$ is rainfall depth at time $t$, and $N_i$ is the drops
number in class $i$. The instrument sums up the number of raindrops in each sampling class and
produces the raindrop spectra for a time step. Here, we use the term $ke$ to represent the
disdrometer-based KE estimated by DSD measured directly every minute. The terminal velocity





of a raindrop can be estimated from its power law empirical relationship with raindrop diameter
(Atlas and Ulbrich, 1977), with this considered more suitable for Chilbolton in the UK (Islam et
al., 2012):

$$v_{At1} = 3.78 D_i^{0.67} \tag{4}$$

Thus, unit rainfall KE estimates per minute are obtained by replacing $v_i$ in Eq. (2) with $v_{Atl}$.
The other form of rainfall KE is expressed at an event scale and represents the sum of the
storm energy covering all time steps covering an event. The individual event energy (MJ ha$^{-1}$) is
calculated as follows:

$$E = \sum_{t=1}^{nt} ke_t P_t \tag{5}$$

where $P_t$ is the rainfall amount (mm) in the $t^{th}$ minute and $nt$ is the time steps number. Historical
rainfall data are divided into wet and dry periods. A string of erosive rainfall storms are first
extracted through the predefined rules. A continuous 6-h dry period interval was used to divide
rainfall events (Hanel et al., 2016), following the "minimum dry-period duration" definition of a
rainfall event (Bonta, 2004). Moreover, a rainfall amount of 12.7 mm was set as the threshold to
filter effective rainfall events (Renard et al., 1997).
Rainfall KE is obtained for a given site based on size and velocity of raindrops. When
disdrometer data are absence, energy can be estimated from empirical relationships using rainfall
intensity $I$ (mm). Five commonly used functions (including exponential, logarithmic, power law,
and inverse proportion) have been mentioned in Section 1. Taking the exponential form as an
example, the rainfall KE at any location can be estimated as:

$$E_{max} = e_{max}(1 - ae^{-bI}) \tag{6}$$

where $e_{max}$ is the mean maximal value of energy measured under high rainfall intensity, and $a$
and $b$ are coefficients modeling the equation curve. Here, minimum KE can be determined by
parameters $a$ and $e_{max}$ together, while the overall shape of the curve is modeled by parameter $b$.





2.2 WRF-based rainfall KE estimation

Differing from disdrometer measurements, the complete DSD cannot be obtained from

the WRF model. Instead, the DSD of the microphysical parameterization (MP) scheme is
handled with a constrained-gamma distribution model, which is defined as:

$$N(D) \ = \ N_0 D^\mu e^{-\lambda D} \tag{7}$$

where $N_0$, $\mu$, and $\lambda$ are the intercept, shape, and slope parameters of the DSD. In terms of double-
moment bulk schemes, $N_0$ and $\lambda$ can be abstracted from the number concentration $N$ and
predicted mixing ratio $q$, as shown below:

$$N_0 \ = \ \frac{N\lambda^{u+1}}{\Gamma(\mu \ + \ 1)} \tag{8}$$

$$\lambda \ = \ \left[ \frac{cN\Gamma(\mu \ + \ d \ + \ 1)}{q\Gamma(\mu \ + \ 1)} \right]^{\frac{1}{d}} \tag{9}$$

$c$ and $d$ are the assumed power-law coefficients between diameter and mass ($m \ = \ cD^d$), and $\Gamma$
represents the function in gamma form  (Morrison et al., 2009). The value of the shape parameter
$\mu$ ($\mu \ = \ 0$) in double-moment schemes is fixed, except for the WRF double-moment 6-class
(WDM6) schemes, following gamma distribution which defined $\mu \ = \ 1$ (Jung et al., 2010;
Johnson et al., 2016).

Because DSD retrieval is sensitive to MPs (Cintineo et al., 2014; Morrison et al., 2015),

the WRF model this study adopted completely or partially three types of double-moment cloud
MP schemes. The Morrison double-moment scheme involves the number concentrations and
mixing ratios of multiple hydrometeors (Morrison et al., 2009). Moreover, the WDM6 scheme
further considers a prognostic factor to estimate and predict the cloud condensation nuclei (CCN)
number concentration (Hong et al., 2010; Lim and Hong, 2010). Finally, the Thompson aerosol-
aware (TAA) scheme can predict both ice nuclei (IN) and CNN number concentrations
(Thompson and Eidhammer, 2014).

The DSD parameters were thus obtained under the three WRF MPs. For theoretical DSD,

*ke* estimates per minute were obtained by integration of the full raindrop size spectrum using:


$$ke'_t = \frac{1}{AR_t} \int_0^\infty N(D) \frac{1}{12} 10^{-6} \pi \rho v_i^2 D_i^3 dD \tag{10}$$

For the WRF-derived DSD covering the whole study area, there was no need to construct a $ke$–I relationship to interpolate KE in ungauged areas. The WRF-based rainfall KE under storm event scale is thus given as:

$$E_W = \sum_{t=1}^{nt} ke'_t \, P_t \tag{11}$$

2.3 Rainfall erosivity estimation

Most storm events have relatively low intensities and KEs with occasional peaks, based on the disdrometer DSD data used to evaluate the rainfall $ke$–I function. Proper estimation of rainfall erosivity potential should consider total KE over a long period. The rainfall erosivity factor (or R-factor) is calculated by a multi-annual average of the total storm erosivity index (Wischmeier and Smith, 1958; Van Dijk et al., 2002), while annual rainfall erosivity $R$ can be obtained using:

$$R = \sum_{m=1}^{M} (EI_{30})_m \tag{12}$$

where $M$ is the total number of erosive events within a year. $(EI_{30})_m$ are total rainfall kinetic energy and maximum 30-min rainfall intensity recorded within 30 consecutive minutes (unit: mm h$^{-1}$), respectively, for the $m^{\text{th}}$ event.

Wischmeier and Smith (1958) first proposed the use of $EI_{30}$, as the rainfall erosivity for each event, based on research data from many sources. $I_{30}$ was calculated to have higher relevance to soil erosion than maximum 5-min, 15-min, or 60-min rainfall intensities (Wischmeier and Smith, 1958). The calculation of $EI_{30}$ initially uses recording-rain gauge data to divide continuous rainfall into time periods with equal rainfall intensity. Because rainfall measurements with high temporal resolutions are required but difficult to obtain from general rainfall measurements, short time equal-interval rainfall data with higher accuracy over multiple years are preferred for estimating $EI_{30}$. For example, Xie et al. (2016) used 1-min rainfall data instead of recording-rain gauge records. For coarse-resolution, equally spaced data, researchers





have proposed a conversion factor to reduce bias error (Weiss, 1964; Williams and Sheridan,
1991).

The rainfall erosivity can be derived from rainfall KE. It plays a main dynamic role in
USLE/RUSLE, representing the potential for soil erosion caused by rainfall. To distinguish the
disdrometer- and WRF-derived rainfall erosivity in this study, we use the terms $R_D$ and $R_W$,
respectively.

### 2.4 Evaluation methods

Because there is no direct way to measure rainfall erosivity across a large area, it is
difficult to validate outcomes using observations. However, $R_D$ is considered to be relatively
accurate due to its specific measurement of raindrops. We therefore assumed that $R_W$ values were
accurate if it closely matched $R_D$ of a given location. A long-term comparison of $R_W$ and $R_D$ at
disdrometer stations was thus conducted to evaluate the validity of $R_W$.
Three indicators were introduced for the evaluation: Pearson's correlation coefficient,
mean absolute error (MAE), and coefficient of determination ($R^2$) (Borrelli et al., 2017). Pearson
correlation coefficient is an index used to evaluate the linear correlation between two variables,
and is defined as follows:

$$Pearson = \frac{n \sum R_{D_i} \sum R_{W_i} - \sum R_{D_i} \sum R_{W_i}}{\sqrt{n \sum R_{D_i}^2 - (\sum R_{D_i})^2} \sqrt{n \sum R_{W_i}^2 - (\sum R_{W_i})^2}} \tag{13}$$

where $n$ is the number of variable samples. Because this correlation cannot reveal the absolute
bias of rainfall erosivity values, the MAE was also used; this is defined as:

$$MAE = \frac{\sum |R_{W_i} - R_{D_i}|}{n} \tag{14}$$

$R^2$ is an indicator to assess the fit of the trend line, expressed as the ratio of the variance
in the dependent variable predicted from the independent variable. It measures the extent to
which the model replicates observations based on the proportion of the results interpreted by the
model to the total change, written as:



$$R^2 = 1 - \frac{SS_{res}}{SS_{tot}} \tag{15}$$

where $SS_{res}$ is the sum of squares of residuals between two variables and $SS_{tot}$ is the total sum of
squares.

**3 Study area and data sources**

The whole of the UK was set as the experimental area for investigating rainfall erosivity
estimation. The UK consists of mostly lowland terrain, with a maximum elevation of 1345 m.
Water and wind are most significant forces of soil erosion in the UK, and together cause
approximately 2.2 million tons of topsoil to be eroded annually, seriously affecting soil
productivity, water quality, and aquatic ecosystems through siltation of watercourses (EA, 2004).
According to the Environmental Agency, the total cost of soil erosion in the UK is approximately
$88 million each year, including an agricultural production loss of $17.6 million (O'Neill, 2007).
More importantly, the changing climate may exacerbate the degree of erosion. For example,
hotter, drier climates make soils more susceptible to wind erosion, and intense storms increase
rainfall erosivity (Defra, 2009). Studies of water erosion in England and Wales (Morgan, 1985;
Evans, 1990) have found that loose soils (especially sand), such as the soils found in Shropshire
and Herefordshire in Wales, are more susceptible to water erosion. In a study of rainfall erosion
in Europe, Panagos et al. (2015a) found that the humid Atlantic climate results in highly variable
rainfall erosivity, such as higher R-factor values in western England and lower values in the
eastern UK.
The gauge datasets used are from the land surface and marine surface measurements
datasets (data availability: 1853–present) provide by the UK Met Office. A network of rain
gauges covering 304 stations across the whole UK observes continuous rainfall data in hours
(Figure 1). The base data of most stations comprises the times of each tip (0.2 mm per tip),
converted into 1-h rain accumulations. The rainfall observations are not always valid for each
hour at each station. The hourly grid-based rainfall maps are then calculated based on ordinary
kriging interpolation of rain gauge network data to obtain the spatial distribution of rainfall for
each time step, as inputs for rainfall erosivity estimation. This wide-range-use geostatistical
approach can account for both the distance and pairwise spatial relationship between points


through variograms. The precipitation interpolation method uses sample gauge points taken at
different locations and creates a continuous surface to achieve an accurate spatial variation
estimation of rainfall patterns.
We used data from two disdrometers in southern England. The first was Chilbolton
station (51 ̊08'N, 1 ̊26'W), with an impact-type Joss–Waldvogel disdrometer (JWD) mainly
used to compute rainfall erosivity. It can measure drop sizes from 0.3 to 5.0 mm in 127 bins. The
sampling period and collector area were 10 s and 50 cm$^2$, respectively. Data were available for
April 2003 to July 2018. The second was the University of Bristol station (51 ̊27'N, 2 ̊36'W),
with an OTT Parsivel$^2$ disdrometer (OPD). Data were available for November 2015 to December
2018. This disdrometer subdivides particles into appropriate classes and has a nominal cross-
sectional area of 54 cm$^2$. The 10-s period measurement data from the two disdrometers were
averaged into a 1-min period to filter out time variations (Montopoli et al., 2008; Islam et al.,
2012; Song et al., 2017).
Meteorological data comes from the ERA-Interim dataset, a global atmosphere re-
analysis product, generated by the ECMWF. For the scientific community, ERA-Interim is
considered to be one of the most important atmospheric datasets, with its data rich period
available since 1979 and updated in current time (Dee et al., 2011). The Integrated Forecasting
System released in 2006 contains a 12-h analysis window derived 4-D variational analysis,
driving the data assimilation system to generate ERA-Interim. The dataset covers 60 vertical
classes of approximately 80 km from the ground to 0.1 hPa. The Gridded Binary format is used
to store data for three months in a separate file. A data processing scheme was established to
collect and retrieve ERA-Interim data of each rainfall event.
The rain gauge and Chilbolton disdrometer datasets can be obtained from British
Atmospheric Data Centre in National Centre for Atmospheric Science research center (MO,
2012). ERA-Interim data can be obtained from the ECMWF Public Dataset website
(https://apps.ecmwf.int/). Considering the availability of the above datasets and model
requirements, we mainly used data covering the period 2004–2017.



## 4 Results

### 4.1 Empirically derived rainfall erosivity estimation

To evaluate the $R_W$, the raindrop spectrum collected by the Chilbolton station disdrometer is used to estimate rainfall KE first. The key in estimating rainfall KE by disdrometer lies on building an empirical relationship between rainfall amount and KE. We used DSD measurements from 2004 to 2013 to establish five empirical relationships between unit rainfall kinetic energy ($ke$) and intensity ($I$) (Table 1), and used 2014–2017 data for the cross validation. It can be seen from Table 1 that the inverse proportional relationship (Equation III in Table 1) had the worst performance, in that both the calibration and validation $R^2$ values were $< 0.3$. The values of the other equations were $> 0.48$, among which the exponential formula (Equation I in Table 1) had the highest calibration $R^2$ (0.50) and validation $R^2$ (0.45), respectively. In addition, the power law formula (Equation V in Table 1) showed a similar performance to the exponential formula at rainfall intensities $< 5$ mm h$^{-1}$. However, the power law formula also had a continuous increasing trend, which may not be suitable for high-intensities. Figure 2 shows the fitted relationship of $ke$–$I$ based on exponential regression. The exponent-based relationship is widely used in the literature and in forecast models such as RUSLE (Renard et al., 1997). We therefore adopted it here as the empirical formula to estimate rainfall erosivity in the UK.

Based on rainfall KE, the point $R_D$ can be obtained at a disdrometer location. In current study, we established a method to estimate the R-factor using 60-min rainfall data. $EI_{30}$ obtained from 1-min DSD data was considered as the standard R-factor at Chilbolton Station. Hourly rain gauge data at the same location were used to calculate $(EI_{30})_{60}$, which refers to $EI_{30}$ calculated from 60-min data. The regression relationship between $EI_{30}$ and $(EI_{30})_{60}$ was then established. The $(EI_{30})_{60}$ of each month, obtained from the 60-min rainfall data of the Chilbolton Station rain gauge in 2004–2013, was calculated. The regression relationship between the monthly sum of $(EI_{30})_{60}$ and the standard monthly $EI_{30}$ from DSD was calculated to obtain a coefficient of 1.836. Rainfall erosivity can subsequently be calculated by multiplying $(EI_{30})_{60}$ by the coefficient.

Beyond assuming that the disdrometer-derived $ke$–$I$ relationship can be applied to a whole study area; point rainfall measurements must be interpolated to obtain areal rainfall values in traditional rainfall erosivity estimation. We obtained 60-min rainfall data from 304 rain gauges around the UK from 2004 to 2017. Note that not all rain gauges were available for the whole


period (available gauges each year are indicated in Figure 3). We used the ordinary kriging
interpolation method to obtain the spatial distribution of rainfall for each time step. This wide-
range-use geostatistical approach can account for both the distance and pairwise spatial
relationship between points through variograms. Figure 3 shows the results of annual rainfall
(*Rain*), annual rainfall kinetic energy (*E*), and annual rainfall erosivity (*R*) for different years.
The distribution trends of *Rain*, *E*, and *R* were similar, and were positively correlated except for
certain locations or periods. For instance, in 2013, *Rain* in the northwestern UK decreased from
west to east, while E and R-factor decreased from south to north; furthermore, areas with large *E*
and *R* values in southeastern UK could not be directly observed from the rain map.
The key concern in traditional rainfall erosivity estimation is the spatial predictability of
the *ke–I* relationship. To verify the regional reliability of this relationship, we used data from a
newer disdrometer located at the University of Bristol, approximately 87 km from Chilbolton
Station. The validation data at Bristol Station discontinuously covered the period 2016–2019.
Figure 4 shows the exponential relationship of *ke–I* at Bristol station, which differed
substantially from that based on data from Chilbolton station. A comparison of the modeled and
observed event rainfall erosivity is shown in Figure 5. The modeled erosivity of rainfall event
was not consistent with the observed event rainfall erosivity. The linear regression coefficient
between these values was > 1.2, which was the result of the low *ke* for Bristol Station, and $R^2$
was < 0.85, indicating large uncertainty associated with disdrometer-based rainfall erosivity
estimation.
In summary, the point rainfall erosivity estimated by disdrometer is considered to be
accurate compared to other methods. However, a large-scaled rainfall erosivity through a simple
interpolation of rainfall KE is subjected to a large uncertainty. In the following analysis, the
point $R_D$ is used to appraise the performance of proposed WRF-based estimated method, and the
$R_D$ in the whole UK is only be used for a general comparison of spatial and temporal distribution
of rainfall erosivity.
4.2 Rainfall and DSD estimation by WRF
We used the WRF model ver. 3.8, which has an Advanced Research WRF dynamical
core, to downscale the ERA-Interim reanalysis data. The double-nested domain configuration
used in the WRF model was centered at 55 °19'N, 2 °21'W and applied at a downscaling ratio of





1:5, a finest grid of 5 km, and a temporal resolution of 1 h. Table 2 lists the detailed parameters
used in this domain configuration. With the top pressure level set at 50 hPa in each, both
domains include 28 vertical levels. To obtain favorable initial weather conditions, the model ran
continuously to obtain five years of WRF simulation results.

Simulations were performed using three different bulk double moment MPs: the

Morrison (Morrison et al., 2009), WDM6 (Hong et al., 2010; Lim and Hong, 2010) and TAA
(Thompson and Eidhammer, 2014) schemes. All three can predict the number concentration and
hydrometeors mixing ratio each time step. The WDM6 scheme also predicts the number
concentration of CCN (Hong et al., 2010; Lim and Hong, 2010), while the TAA scheme are able
to predict both IN and CCN number concentrations (Thompson and Eidhammer, 2014).
Additionally, other physical parameterizations include the Dudhia shortwave radiation scheme
(Dudhia, 1989), Mellor–Yamada–Janjic planetary boundary layer scheme (Janjić, 1994), RRTM
longwave radiation scheme (Mlawer et al., 1997), the Noah land-surface model (Ek et al., 2003),
and the Kain–Fritsch cumulus scheme (Kain, 2004),.

The median volume diameter parameter ($D_0$) and generalized intercept parameter ($N_w$)

are generally used in DSD model of WRF (Islam et al., 2012).

$$N_W = \frac{N_0 D_m{}^\mu}{f(\mu)} \tag{16}$$

$$f(\mu) = \frac{6(4 + \mu)^{\mu+4}}{4^4 \Gamma(\mu + 4)} \tag{17}$$

where $D_m$ is the mass-weighted mean diameter. The $f(\mu)$ is a function of the shape parameter $\mu$.
The parameter $\mu$ is assumed as zero or one (based on microphysical scheme configuration) in
WRF. Figure 6 displays the spatial distribution of $D_0$ and generalized intercept parameter $N_w$ for
a given day with rainfall countrywide (January 10, 2013). $D_0$ and $N_w$ had similar patterns, and
were mainly distributed across the southwestern and northeastern UK. The white strip in the
middle of Figure 6 represents an area that received no rain. However, the three MPs yielded large
differences; $D_0$ of MP-TAA was the highest among three MPs, whereas $N_w$ of MP-WDM6 was
much larger than others. In addition, $D_0$ and $N_w$ did not consistently show a positive correlation.





The different MP estimation results underscore the complexity of the rainfall process, which is
the reason we estimated rainfall KE using WRF schemes instead of traditional formulas.

4.3 Comparison of WRF- and disdrometer-derived rainfall erosivity at Chilbolton station

With the WRF-based rainfall intensity and DSD estimations, rainfall erosivity was

derived using Equations (10)–(12). Hereafter, this is referred to as $R_W$, which is further
distinguished based on the three MP schemes used: $R_{W-Morrison}$, $R_{W-WDM6}$, and $R_{W-TAA}$. Figure 7
compares disdrometer- and WRF-derived monthly rainfall erosivity estimations at Chilbolton
Station for the period 2014–2017. The general patterns of the four rainfall erosivity values were
similar. $R_{W-Morrison}$ tended to be larger than $R_D$ in some months, whereas $R_{W-TAA}$ matched the $R_D$
value relatively well for smaller values. Because WRF data were taken from a $2 \times 2$-km grid
around Chilbolton Station, there was spatial error in addition to the systematic error of estimating
rainfall erosivity.

Table 3 shows the correlation indicator results between $R_D$ and the three type $R_W$ at

Chilbolton station. The Pearson correlation coefficients generally exceeded 0.7, supporting the
potential utility of WRF-based estimation. In terms of MAE, $R_{W-TAA}$ had the best performance
(6.51), whereas $R_{W-Morrison}$ and $R_{W-WDM6}$ showed slightly worse performance (approximately 8).
Among the three schemes, $R_{W-TAA}$ had the best fit with $R_D$. The indicators and comparison results
suggest that the deviations in results need to be considered; a method of bias elimination is
therefore described in Section 4.4.

4.4 $R_W$ estimation for the whole UK

The $R_W$ at Chilbolton station showed obvious systematic deviations compared with the

disdrometer-derived results (see Section 4.2 and 4.3). A simple bias correction was therefore
applied to adjust the individual storm KE estimations of $R_W$. The biases from dividing average
$R_{W-Morrison}$, $R_{W-WDM6}$, and $R_{W-TAA}$ by average $R_D$ during 2014-2017 were 0.55, 0.20, and 0.36,
respectively.

The rainfall erosivity distribution for the whole UK was then obtained. Figure 8 shows

the distribution of $R_W$ at the annual scale covering the period 2013–2017. The pattern of the
rainfall erosivity maps showed a general regional-dominant characteristic. For example, it
always decreased from west to east, predominantly shaped by orography. Affected by the





prevailing westerly winds, there was abundant rainfall in the western and northern mountains, as
indicated by high rainfall KE values in these regions. In addition, among the study years, 2014
and 2015 showed higher national rainfall erosivity, with a large range in the west coast area.
Figure 9 shows the average $R$ distribution for 2013–2017 estimated by rain gauges and
WRF MPs. WRF grids could cover all regions in the UK evenly, offering more detailed erosivity
results, especially in the mountainous northwestern region. Here, values of average $R$ map
calculated by rain gauges were much higher than three type $R_W$, although they all have $R$
decreased from west to east. Noted that $ke$–$I$ empirical equation at Chilbolton station used in the
whole UK, will not always be accurate in regions with different rainfall characteristics. In terms
of $R_W$ results, the three MPs obtained the same spatial pattern in rainfall erosivity, where $R_{W\text{-}WDM6}$
yielded the greatest geographical difference. It is clear that the proposed WRF-based estimated
method can capture more details of the spatial change of rainfall erosivity compares with the
traditional disdrometer-based method.
To evaluate the change in rainfall erosivity with time in the UK, the average value of all
the WRF grids covering the whole UK was calculated over 2013–2017 (Figure 10). The average
$R_W$ trends of $R_{W\text{-}Morrison}$ and $R_{W\text{-}TAA}$ were similar, both increasing from a minimum in 2013 to a
maximum in 2014, and then gradually decreasing from 2014 to 2017. The red line in Figure 10
indicates a series of mean values of the three MPs results, which varied from 36,782 to 51,600
MJ mm ha$^{-1}$ h$^{-1}$ y$^{-1}$ (mean: 43,216 MJ mm ha$^{-1}$ h$^{-1}$ y$^{-1}$).
The maximum values for $R_{W\text{-}Morrison}$ and $R_{W\text{-}TAA}$ occurred in 2014, whereas that of $R_{W\text{-}WDM6}$
occurred in 2015. A sequence of extreme weather events occurred in the UK in 2014, including
major winter storms in late January to mid-February, which caused widespread flooding and
other economic losses, and greatly increased rainfall erosivity that year. However, the gauge-
based interpolation map shows the average annual rainfall amount for the years 2013–2017 were
884.9, 1014.0, 1008.5, 894.9, and 937.3 mm, respectively. The large rainfall erosivity difference
between 2014 and 2015, and the two years with similar rainfall amount, indicates that much
rainfall erosion occurs during the rainfall events of high intensity instead of simply high rainfall
amount. More notable variation pattern of rainfall erosivity may be found with longer simulation.
The strength of the proposed method lies on its ability to estimate large covering and long-term
rainfall erosivity.





## 5 Conclusions

This study presented a novel method for large-scale rainfall KE and erosivity estimation based on high resolution WRF-derived DSDs. Three microphysical parameterizations schemes (Morrison, WDM6, and Thompson aerosol-aware [TAA]) were designed to obtain raindrop size distributions, rainfall KE and rainfall erosivity at the whole UK scale covering the period of 2013-2017. With validation by the long-term observations of a disdrometer, the WRF-based rainfall erosivity showed acceptable performance at Chilbolton station. Among the three WRF schemes, TAA performed best and was recommended for the future investigation. The results revealed that high rainfall erosivity occurred in the west coast area in the UK. Compared with the traditional empirical method, the proposed method can explain rainfall erosivity from a microphysical perspective, and reflect more spatial variation due to changes in rainfall KE at the whole-country scale. The development of a numerical weather prediction model therefore offers an opportunity to better understand rainfall erosivity directly from its true definition. More importantly, because the WRF model is able to be driven by the global reanalysis data to obtain large-scale rainfall kinetic information, the proposed scheme can be easily applied to other regions, especially in ungauged areas.

Although an acceptable rainfall erosivity estimation is obtained using the WRF model, some uncertainties associated with it cannot be ignored. For example, as the MPs of WRF were closely related to DSD, improper determination of MPs will introduce additional uncertainty. The marked discrepancy among the three schemes (especially between Morrison and the others) in this study underscored the possible uncertainty associated with $R_W$. Moreover, the measurement error by disdrometer may also contaminate the evaluation process. For example, when comparing the observed raindrop velocities based on the disdrometer at Bristol station with their empirical values, we observed a dispersion of raindrops, with a number of drops showing significant deviations. This velocity distribution resulted in an uncertainty in $ke$ estimation.

In addition, other sources of uncertainty, such as temporal downscaling of rainfall and point-to-area representative error by WRF, may introduce further uncertainty, which should be put in perspective of future work. It is expected that more exploration of research areas with different climatic and geographical characteristics would help us to establish a greater degree of accuracy on this matter.



**Acknowledgments**

This work was supported by the National Natural Science Foundation of China (Nos. 41871299 and 41771424), and the National Key R & D Program of China (Nos. 2018YFB0505500, 2018YFB0505502). The authors acknowledge the British Atmospheric Data Centre and the European Centre for Medium-range Weather Forecasts as the sources of data used in the study.

The rain gauge datasets and Chilbolton disdrometers were sourced from the Met Office Integrated Data Archive System (MIDAS). Both datasets are available from the NCAS British Atmospheric Data Centre (http://archive.ceda.ac.uk/). The ERA-Interim data driving the WRF model can be downloaded from the ECMWF Public Datasets web interface (https://www.ecmwf.int/).

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

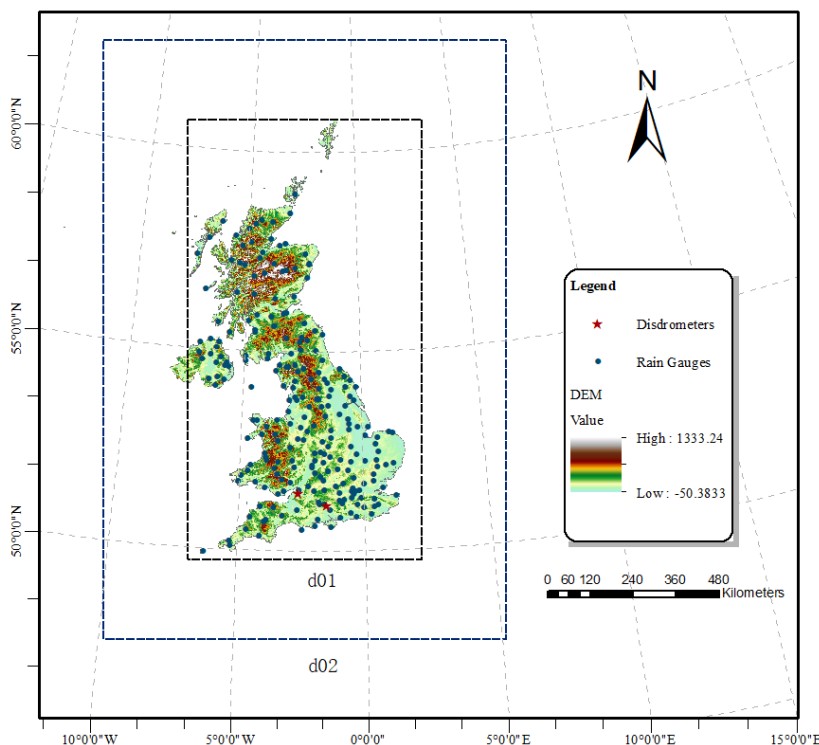


**Figure 1**. Location of rain gauges, Joss–Waldvogel disdrometer (JWD) at Chilbolton
Observatory, OTT Parsivel[2] disdrometer (OPD) at Bristol Observatory and configurations of
domain setups in the WRF model.






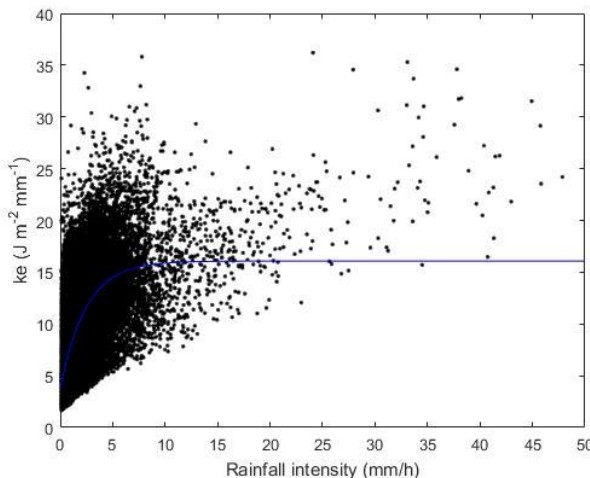


**Figure 2**. The fitted relationship of *ke–I* based on exponential regression (2004–2013).


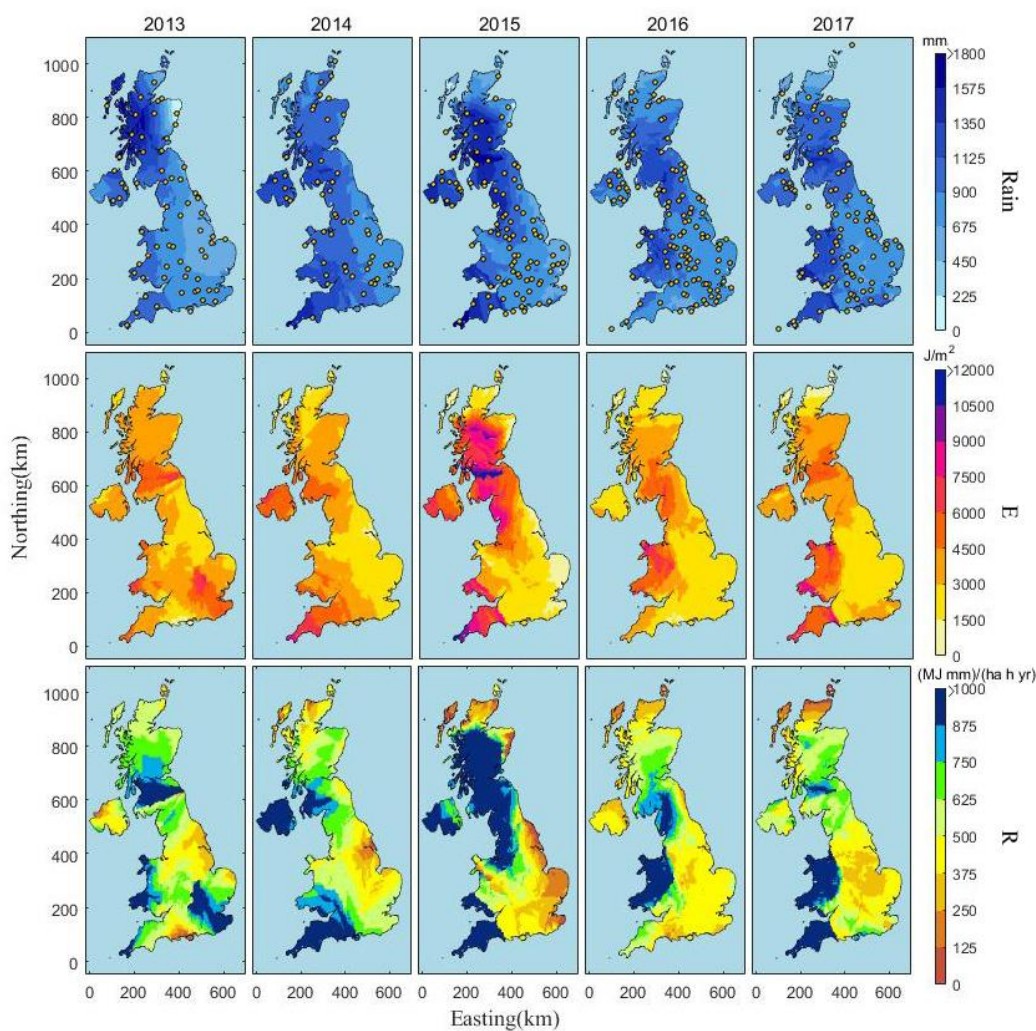

**Figure 3**. Gauge-based interpolation maps of annual rainfall amount (*Rain*), rainfall kinetic energy (*E*) and rainfall erosivity (*R*) in 2013-2017.





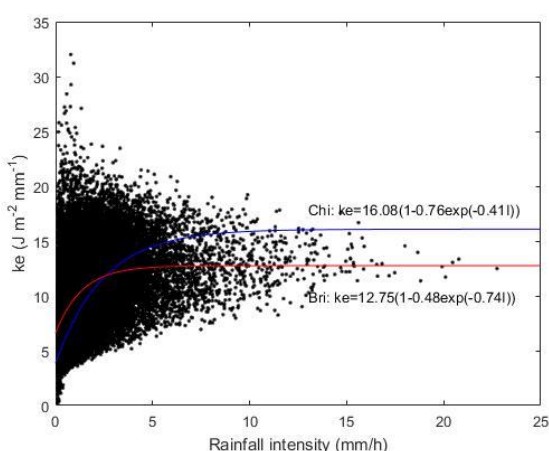


**Figure 4**. Relationship of *ke–I* at Bristol station.






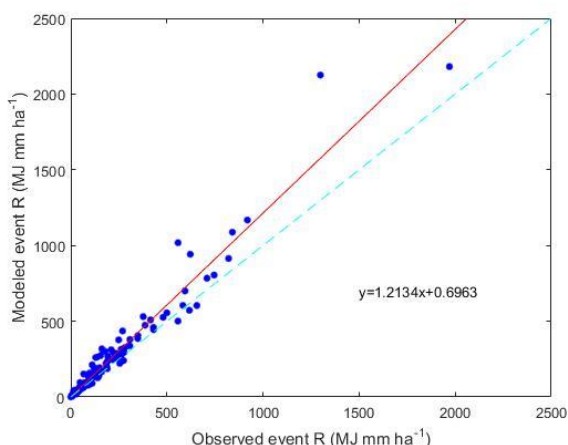


**Figure 5**. Comparison of observed and modeled event rainfall erosivity covering the period of
2016–2019.




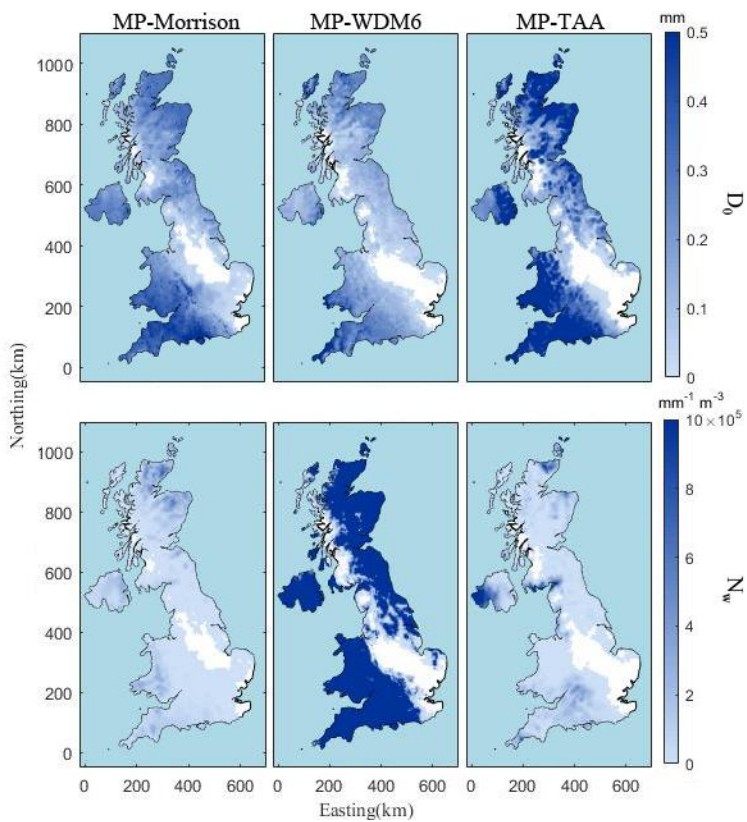

**Figure 6**. Map of average WRF DSD $D_0$ and $N_w$ (January 10, 2013).



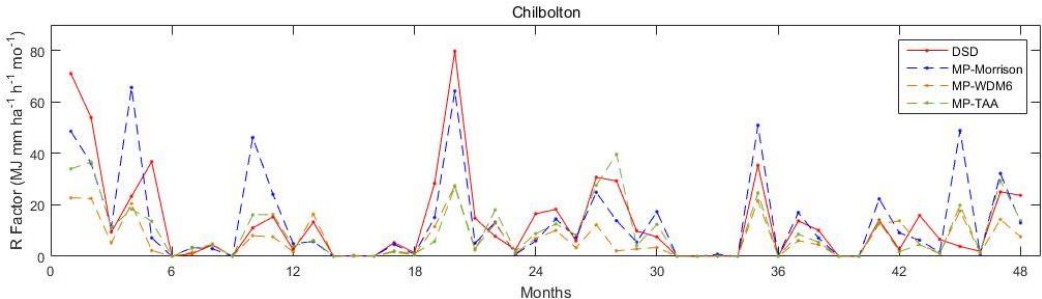


**Figure 7**. Comparison of disdrometer- and WRF-derived monthly rainfall erosivity estimations

at Chilbolton station.


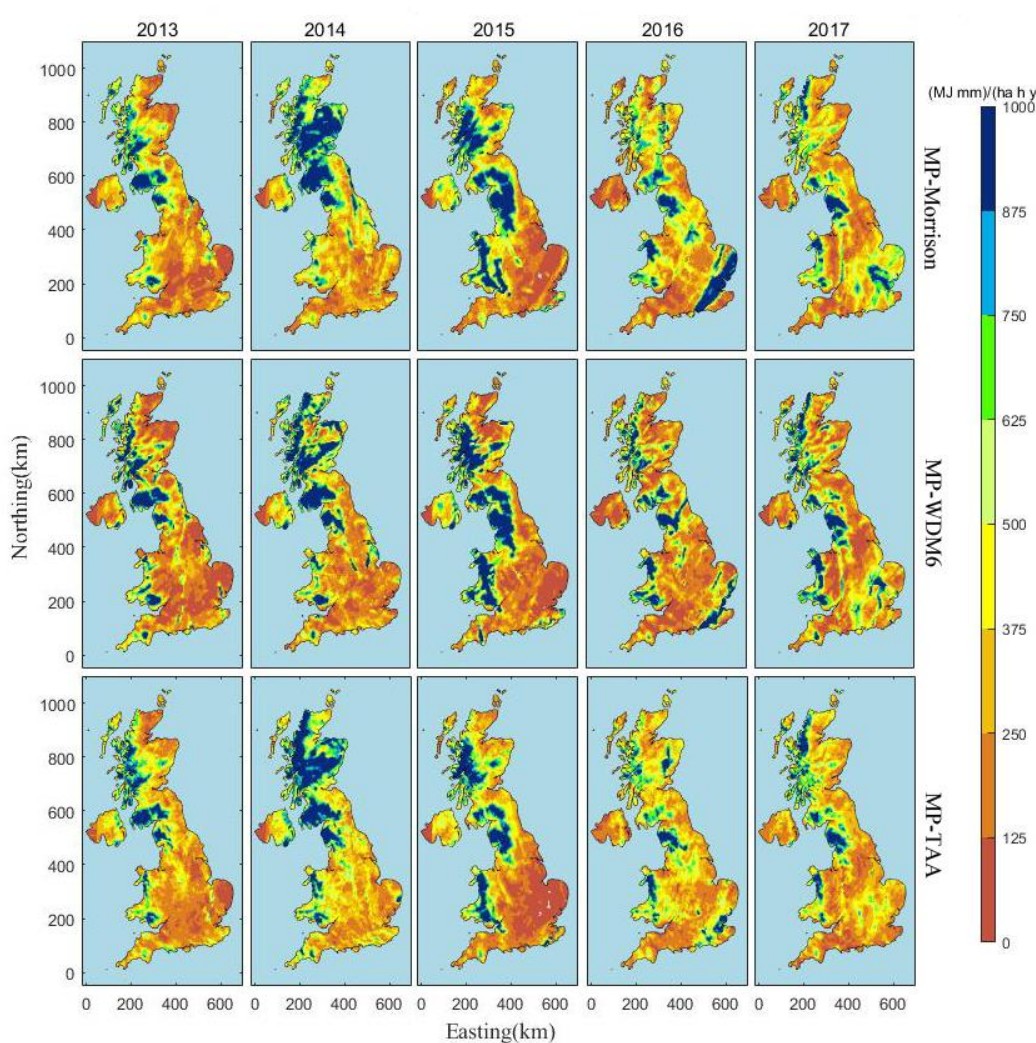

**Figure 8**. $R_W$ maps of the whole UK for different years.

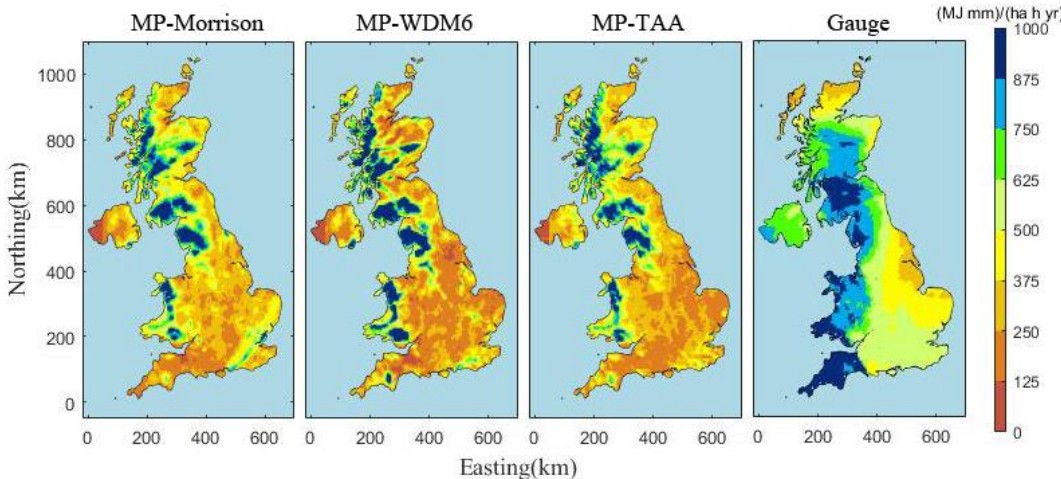


**Figure 9**. The 5-year (2013–2017) average *R* maps based on WRF grids and rain gauge

interpolation.





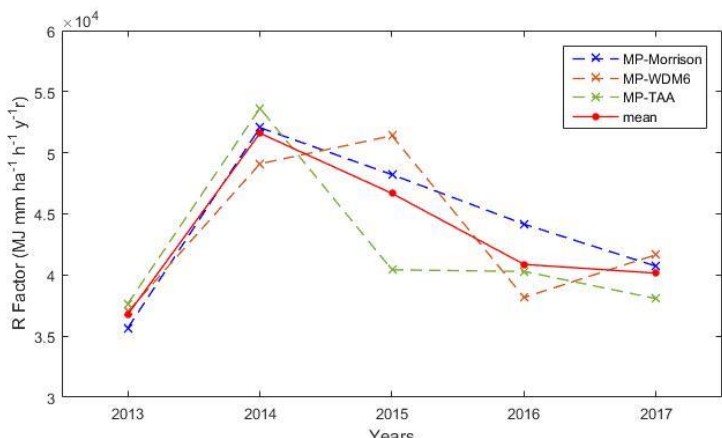


**Figure 10**. The average $R_W$ of all the WRF grids covering the whole UK (2013–2017).






**Table 1.** Relationship of *ke–I* at Chilbolton Station.

| ID | Equation | Calibration $R^2$ | Validation $R^2$ |
|:--:|:--:|:--:|:--:|
| I | $ke = 16.08(1 - 0.76e^{-0.41I})$ | 0.50 | 0.45 |
| II | $ke = 8.65 + 6.39\lg(I)$ | 0.48 | 0.43 |
| III | $ke = 10.19 - 1.05/I$ | 0.29 | 0.25 |
| IV | $ke = 8.65 + 2.78\ln(I)$ | 0.48 | 0.43 |
| V | $ke = 8.12I^{0.34}$ | 0.50 | 0.45 |







**Table 2.** The configurations of WRF model for two nested domains.

| Domain | Domain size (km) | Grid Spacing (km) | Grid size | Downscaling ratio |
|--------|------------------|-------------------|-----------|-------------------|
| d01 | $1,125 \times 1,675$ | 25 | $45 \times 67$ | - |
| d02 | $655 \times 1,230$ | 5 | $131 \times 246$ | 1:5 |







**Table 3**. Indicators comparison between $R_D$ and three type $R_W$ at Chilbolton station on monthly

scale.

| Indicators | MP-Morrison | MP-WDM6 | MP-TAA |
|---|---|---|---|
| Pearson | 0.71 | 0.77 | 0.79 |
| MAE | 8.05 | 8.42 | 6.51 |
| $R^2$ | 0.42 | 0.31 | 0.54 |