# Peer review of "Estimation of rainfall erosivity based on WRF-derived raindrop size distributions"

_Hydrology and Earth System Sciences, 2020_

## Referee Comment (RC1) · Anonymous Referee #1 · 12 Jul 2020

This study presents a novel approach to estimate the rainfall erosivity based on WRF products, which provides a new insight in describing the rainfall erosivity in a large scale. The approach is the first attempt and it is understandable that further exploration is required. However, here are some points that needed to be illustrated for this manuscript. - Statement of the third key point is not very clear. After reading the manuscript, I know the main point is the west coastal area, but the statement is not emphasizing this. - For interpolation of rainfall in section 4.1, CEH also published 1km gridded rainfall datasets for the whole UK, have you compared your interpolation rainfall with theirs? The reason I'm asking it is because rainfall interpolation is important in the following analysis of erosion, it's worthy to ensure that the interpolation is reliable. - The empirical equation in table 1 and figure 1 did not perform very well with R2 not over

0.50, how well is the relationship in other studies? Is this acceptable based on previous studies? - The two disdrometers are located in the same region, but the relationship is significantly different. Is it common in previous studies or any explanation about it? - In figure 7, can you change the x axis tick to the real month, e.g. Jan/2013, so that seasonable patterns can be observed and analyzed? - Discussion part is weak in the manuscript, more discussions can be added in the result section or a separate section by comparing with previous studies and discussing about the potential limitations and applications of this approach.

―――――――――――――――――――――――

---

## Referee Comment (RC2) · Anonymous Referee #2 · 20 Jul 2020

This paper proposes a new approach to estimating rainfall erosivity based on Weather Research and Forecasting model. This study offers new insights on determining the rainfall-driven erosion in regional scale (which is the UK scale in the study) using a combination of different datasets and techniques. To this end, different products (real-time of continuing rainfall measurements from 304 gauge stations), technology (raindrop size distrometer) and methodology (WRF model) were implemented. However, the study can be further improved if the following points can be considered by the authors: - You have used two distrometers in the same locations (considering the whole UK study area) and in the same elevation ranges (low elevation), but they differ considerably. What about the high elevation then? And how much they are representative of the whole UK?

[Figure]

- Could you use the recently published and open access Distrometer Verification Network of UK (Disdrometer Verification Network (DiVeN): a UK network of laser precipitation instruments, https://amt.copernicus.org/articles/12/5845/2019/) to support the finding of your study and refine better the findings? - The performance (R2) of equations of the relationship between Ke-I presented in Table 1 are low and very similar (except ID-III). The exponential (ID-I) and power-law (ID-V) are exactly the same, and did not support the statement given in Line 73 where the exponential relationship is used in preference. Would you discuss this in detailed and how much these values are in line with former investigations? - The Discussion section is missing. The Discussion section is one of the most exciting parts of any study and preferably to be presented separately from the Results section. Add the Discussion section and compared the study finding with previous studies. - What about ground truthing validation of your results in the whole UK or using previous studies with experimental and in-situ data? - How much your study can be compared or can support the very recently published research entitled "National-scale geodata describe widespread accelerated soil erosion" https://doi.org/10.1016/j.geoderma.2020.114378. The latter publication can enrich the discussion part of the study. Specific comments: - Avoid using the abbreviation in the abstract and key points. - Enrich the Figures and Tables captions, ensuring self-explaining to the readers without referring to the main text and avoiding abbreviations.

---

## Author Comment (AC1) · 10 Aug 2020

Point 1: Statement of the third key point is not very clear. After reading the manuscript, I know the main point is the west coastal area, but the statement is not emphasizing this.

Response 1: Agreed and the following text has been added in the Section 4.4:

"The highest rainfall erosivity regions in the UK are concentrated in the mountainous areas along the western coast, related to their rainfall system. The moist air brought by the prevailing westerly wind from the Atlantic Ocean moves from west to east across the UK and rises when it encounters the mountains of western England. Therefore, the mountainous regions along the UK western coast have the highest rainfall amount and

rainfall erosivity in the UK. In addition, western Scotland is under the subpolar oceanic climate, which enhances its humidity. On the contrary, eastern Scotland and northeastern England are more likely to expose continental polar air mass, which brings dry and cold air and lower rainfall erosivity."

Point 2: For interpolation of rainfall in section 4.1, CEH also published 1km gridded rainfall datasets for the whole UK, have you compared your interpolation rainfall with theirs? The reason I'm asking it is because rainfall interpolation is important in the following analysis of erosion, it's worthy to ensure that the interpolation is reliable.

Response 2: CEH is a dataset with 1 km gridded estimates of daily and monthly rainfall for the whole UK derived from the Met Office. The natural neighbour interpolation methodology, including a normalisation step based on average annual rainfall, was used to generate the product (Tanguy et al., 2019). The method for calculating EI30 requires hyetograph data for individual storms (Wischemier and Smith, 1978). Therefore, the monthly or daily rainfall data generated by CEH are hard to distinguish rainfall events and estimate EI30, although some studies have proposed methods related daily rainfall data to estimate rainfall erosivity using statistical models.

The study area has the little climatic variability with same climate type named temperate oceanic climate, and the 304 hourly rain gauges are distributed throughout the UK evenly. Therefore, ordinary kriging interpolation was expected to produce realistic results. It should be noted that refined interpolation for rain gauges is not the focus of this research. Instead, we tried to propose a methodology based on the numerical weather prediction model for estimating rainfall erosivity anywhere around the world, especially those regions with sparse instruments.

Tanguy, M.; Dixon, H.; Prosdocimi, I.; Morris, D.G.; Keller, V.D.J. (2019). Gridded estimates of daily and monthly areal rainfall for the United Kingdom (1890-2017) [CEH-GEAR]. NERC Environmental Information Data Centre. (Dataset)

Wischmeier, W. H. and Smith, D. D. (1978). Predicting rainfall erosion losses-a guide

to conservation planning. Department of Agriculture, Science and Education Administration, US.

Point 3: The empirical equation in table 1 and figure 1 did not perform very well with R2 not over 0.50, how well is the relationship in other studies? Is this acceptable based on previous studies?

Response 3: The Figure 2 in the manuscript has been replaced with the following figures, which is clearer in showing the method performance. Van Dijk et al. (2002) compared the three forms of ke-I relationships including exponential, logarithmic and power-law equations, based on the same observed data. The R2 values are 0.53, 0.52 and 0.53, respectively. The R2 values in this study is surely acceptable that many studies have obtained R2 value between 0.45-0.50 (Laws and Parsons, 1943; Kinnel, 1980; Brandt, 1988). Angulo-Martínez et al. (2016) compared simulated ke from 14 different exponential ke–I relationships with respect to the disdrometer-observed values, found that R2 was low at 1 min resolution (~0.25). The low R2 of empirical equations also indicate the large variability of DSD in nature. Therefore, we believed that the study of large-scale rainfall energy and rainfall erosivity based on NWP-derived DSD is of great significance.

Angulo-Martínez, M., Beguería, S. and Kyselÿ, J. (2016). Use of disdrometer data to evaluate the relationship of rainfall kinetic energy and intensity (KE-I). Science of the Total Environment 568: 83-94.

Van Dijk, A., Bruijnzeel, L. and Rosewell, C. (2002). Rainfall intensity–kinetic energy relationships: a critical literature appraisal. Journal of Hydrology 261(1-4): 1-23.

Point 4: The two disdrometers are located in the same region, but the relationship is significantly different. Is it common in previous studies or any explanation about it?

Response 4: The current studies have showed that DSD and ke-I relationships changes significantly with geographical locations and weather systems, including climate, altitude and terrain (Van Dijk et al., 2002; Angulo-Martínez et al., 2016). Both disdrometers located in southern England, have the similar oceanic climate. However, there are still differences between the two stations in altitude, topography, land cover, etc. For instance, Chilbolton Observatory is located on the edge of the village of Chilbolton, at an attitude of 86m, while University of Bristol is an urban campus, at 77m attitude. The former is 11 kilometers from the coastline and the latter is above 37 kilometers. From the revised Figure 2 and Figure 4, the difference in the ke-I relationship between the two stations can be clearly observed. Moreover, the significant difference also shows that DSD-based estimation methods are needed to reflect rainfall microphysical characteristics on large-scale, which is the goal of this work.

Van Dijk, A., Bruijnzeel, L. and Rosewell, C. (2002). Rainfall intensity–kinetic energy relationships: a critical literature appraisal. Journal of Hydrology 261(1-4): 1-23.

Point 5: In figure 7, can you change the x axis tick to the real month, e.g. Jan/2013, so that seasonable patterns can be observed and analyzed?

Response 5: Agreed and amended. Figure 7 in manuscript has been changed and the following text has been added in the Section 4.3. We also added a figure (Figure 8) to show how monthly patterns performed.

"Based on the four-year data, the study area is rainy throughout the year with little R monthly, or seasonal patterns change (Figure 8), influenced by the temperate oceanic climate. Figure 8 also indicated that through the perspective of monthly average results, RW-WDM6 values are low, RW-TAA has a good similarity with low RD, and RW-Morrison is the closest to RD in value."

Point 6: Discussion part is weak in the manuscript, more discussions can be added in the result section or a separate section by comparing with previous studies and discussing about the potential limitations and applications of this approach.

Response 6: Discussion part is mainly contained in the conclusion section. The following text has been added in the Section 5 to enrich the discussion:

"The reliability of the WRF model is heavily dependent on the model-driving initial data provided by mesoscale or global models and complicated scheme setting and parameter adjustment (Liu et al., 2013; Thompson and Eidhammer, 2014; Kumar et al., 2017). However, numerous uncertainties are observed in the parameterization of the WRF simulation, and the choice of microphysical schemes has a significant influence on the inverted DSD (Ćurić et al., 2009; Yang et al., 2019). Therefore, combining the DSDs obtained by an increasing number of disdrometers and the WRF model is valuable. For example, the Disdrometer Verification Network (DiVeN) in the UK (Pickering et al., 2019) started in Feb 2017 can be introduced to support and improve our estimation in future studies."

"Soil erosion in the UK is dominated by water erosion (10–30 t km−2 yr−1), especially in areas with abundant rainfall in Scotland, where the soil loss rate is approximately 5–10 times that of dry areas (Duck, 1996). Thus, it is significant to estimate rainfall erosivity to elucidate the microphysical characteristics of rainfall and rainfall–soil interactions. Benaud et al. (2020) collated empirical soil erosion observations from UK-based studies into a geodatabase. However, there is a limitation that this database does not cover the entirety of the UK, especially the limited records in northern Scotland. In our future work, we propose to compare the soil loss database with our estimated soil loss using WRF DSD based rainfall erosivity and a soil erosion model (such as RUSLE). We believe that not only can we better analyze the impact of rainfall and rainfall erosivity on the UK soil loss, but also help to better understand microphysical rainfall–soil interactions to support the rational formulation of soil and water conservation planning."

Benaud, P., Anderson, K., Evans, M., Farrow, L., Glendell, M., James, M. R., ... & Brazier, R. E. (2020). National-scale geodata describe widespread accelerated soil erosion. Geoderma, 371: 114378.

Ćurić, M., Janc, D., Vučković, V. and Kovačević, N. (2009). The impact of the choice of

the entire drop size distribution function on Cumulonimbus characteristics. Meteorologische Zeitschrift 18(2): 207-222.

Duck, R. W. (1996). Regional variations of fluvial sediment yield in eastern Scotland. Erosion and Sediment Yield: Global and Regional Perspectives: Proceedings of an International Symposium Held at Exeter, UK, IAHS.

Kumar, P., Kishtawal, C. and Pal, P. (2017). Impact of ECMWF, NCEP, and NCMRWF global model analysis on the WRF model forecast over Indian Region. Theoretical and Applied Climatology 127(1-2): 143-151.

Liu, J., Bray, M. and Han, D. (2013). Exploring the effect of data assimilation by WRF-3DVar for numerical rainfall prediction with different types of storm events. Hydrological Processes 27(25): 3627-3640.

Pickering, B. S., Neely III, R. R., & Harrison, D. (2019). The Disdrometer Verification Network (DiVeN): a UK network of laser precipitation instruments. Atmospheric Measurement Techniques 12: 5845-5861.

Thompson, G. and Eidhammer, T. (2014). A study of aerosol impacts on clouds and precipitation development in a large winter cyclone. Journal of the Atmospheric Sciences 71(10): 3636-3658.

Yang, Q., Dai, Q., Han, D., Chen, Y. and Zhang, S. (2019). Sensitivity analysis of raindrop size distribution parameterizations in WRF rainfall simulation. Atmospheric Research 228: 1-13.

[Figure]

**Fig. 1.** (new Figure 2 in manuscript). Minutes number per intensity class (x-axis) and ke class (y-axis) with five fitted ke–I curves at Chilbolton station (2004–2013), plotted on linear (left) and logarithmic

[Figure]

**Fig. 2.** (new Figure 4 in manuscript). Minutes number per intensity class (x-axis) and ke class (y-axis) with fitted ke–I curves at Bristol station (2015–2018), plotted on linear (left) and logarithmic (right)

[Figure]

**Fig. 3.** (new Figure 7 in manuscript). Comparison of disdrometer- and WRF-derived monthly rainfall erosivity estimations at Chilbolton station (2014–2017).

[Figure]

**Fig. 4.** (added as Figure 8 in manuscript). Comparison of disdrometer- and WRF-derived average monthly rainfall erosivity estimations at Chilbolton station (2014–2017).

---

## Author Comment (AC2) · 10 Aug 2020

Point 1: You have used two distrometers in the same locations (considering the whole UK study area) and in the same elevation ranges (low elevation), but they differ considerably. What about the high elevation then? And how much they are representative of the whole UK?

Response 1: The current studies showed that DSD and ke-I relationships changes with geographical locations and weather systems, including climate, altitude and terrain (Van Dijk et al., 2002; Angulo-Martínez et al., 2016). Both the two disdrometers located in southern England, have the similar oceanic climate. The focus of this study is not to use disdrometers to estimate rainfall erosivity. On the contrary, we chose the

two disdrometers in similar locations to illustrate the spatial uncertainty of the ke-I relationship exactly. The results indicated that it is inappropriate to rely on an empirical formula in a large scale. The widely used (R)USLE approach to predict ke-I relationships based on measurement at a single location only (Wischmeier et al., 1978; Renard et al., 1997). Therefore, the proposed method based on NWP DSD is expected to effectively improve large-scale rainfall KE and rainfall erosivity estimation.

Angulo-Martínez, M. and Barros, A. (2015). Measurement uncertainty in rainfall kinetic energy and intensity relationships for soil erosion studies: An evaluation using PARSIVEL disdrometers in the Southern Appalachian Mountains. Geomorphology 228: 28-40.

Renard, K. G., Foster, G. R., Weesies, G., McCool, D. and Yoder, D. (1997). Predicting soil erosion by water: a guide to conservation planning with the Revised Universal Soil Loss Equation (RUSLE), United States Department of Agriculture Washington, DC.

Van Dijk, A., Bruijnzeel, L. and Rosewell, C. (2002). Rainfall intensity–kinetic energy relationships: a critical literature appraisal. Journal of Hydrology 261(1-4): 1-23.

Wischmeier, W. H. and Smith, D. D. (1978). Predicting rainfall erosion losses-a guide to conservation planning. Department of Agriculture, Science and Education Administration, US.

Point 2: Could you use the recently published and open access Disdrometer Verification Network of UK (Disdrometer Verification Network (DiVeN): a UK network of laser precipitation instruments, https://amt.copernicus.org/articles/12/5845/2019/) to support the finding of your study and refine better the findings?

Response 2: This study used two disdrometers in Chilbolton and Bristol to calibrate R results derived by WRF model. Both two disdrometers have long running periods and have been fully studied and calibrated from a series of research work by our team (Islam et al., 2012; Dai et al., 2014; Yang et al., 2019). The DiVeN disdrometer network may provide an interesting support for our follow-up research, such as finding an empirical formula that is most suitable for the UK as a whole. However, for this study, DiVeX has less overlap with the period studied here, which has limitations for verifying. The following text has been added at the end of Section 5:

"For example, the Disdrometer Verification Network (DiVeN) in the UK (Pickering et al., 2019) started in Feb 2017 can be introduced to support and improve our estimation in future studies."

Dai, Q. and Han, D. (2014). Exploration of discrepancy between radar and gauge rainfall estimates driven by wind fields. Water Resources Research 50(11): 8571-8588.

Islam, T., Rico-Ramirez, M. A., Thurai, M. and Han, D. (2012). Characteristics of raindrop spectra as normalized gamma distribution from a Joss–Waldvogel disdrometer. Atmospheric Research 108: 57-73.

Pickering, B. S., Neely III, R. R., & Harrison, D. (2019). The Disdrometer Verification Network (DiVeN): a UK network of laser precipitation instruments. Atmospheric Measurement Techniques 12: 5845-5861.

Yang, Q., Dai, Q., Han, D., Chen, Y., and Zhang, S. (2019). Sensitivity analysis of raindrop size distribution parameterizations in weather research and forecasting rainfall simulation. Atmospheric Research 228:1-13

Point 3: The performance ($R^2$) of equations of the relationship between Ke-I presented in Table 1 are low and very similar (except ID-III). The exponential (ID-I) and power-law (ID-V) are exactly the same, and did not support the statement given in Line 73 where the exponential relationship is used in preference. Would you discuss this in detailed and how much these values are in line with former investigations?

Response 3: Figure 1 below replaced Figure 2 in the manuscript, expressed the number of minutes per intensity class (x-axis) and ke class (y-axis). It clearly showed how the five equations performed, plotted on linear and logarithmic intensity scales, respectively. Figure 4 in manuscript also changed to a similar expression. A detailed discussion about the comparison of relationships has been added in section 4.1 as follows:

"Figure 2 shows the ke–I relationship and five fitted curves at Chilbolton station. It can be seen that the two logarithmic curves (Equation II and IV) invariably overlap. The logarithmic form has been used for a long time in USLE (Wischemier and Smith, 1978). It describes ke well at both low and high I, but does not have an upper limit. The power law curve (Equation V) can predict ke well at lower I but overestimates ke at high I. The exponent-based relationship (Equation I) is widely used in the literature and in forecast models such as RUSLE (Renard et al., 1997), which fits the data particularly well in Figure 2. Even though ke in exponential curve has a minimum value at very low I, it also should be noted that higher rainfall intensities are much more important in determining overall storm energy than lower intensities. Therefore, we adopted it here as the empirical formula to estimate rainfall erosivity in the UK."

Point 4: The Discussion section is one of the most exciting parts of any study and preferably to be presented separately from the Results section. Add the Discussion section and compared the study finding with previous studies.

Response 4: Discussion part is mainly contained in the conclusion section. The following text has been added in the Section 5 to enrich the discussion:

"The reliability of the WRF model is heavily dependent on the model-driving initial data provided by mesoscale or global models and complicated scheme setting and parameter adjustment (Liu et al., 2013; Thompson and Eidhammer, 2014; Kumar et al., 2017). However, numerous uncertainties are observed in the parameterization of the WRF simulation, and the choice of microphysical schemes has a significant influence on the inverted DSD (Ćurić et al., 2009; Yang et al., 2019). Therefore, combining the DSDs obtained by an increasing number of disdrometers and the WRF model is valuable. For example, the Disdrometer Verification Network (DiVeN) in the UK (Pickering et al.,

2019) started in Feb 2017 can be introduced to support and improve our estimation in future studies."

"Soil erosion in the UK is dominated by water erosion (10–30 t km$-2$ yr$-1$), especially in areas with abundant rainfall in Scotland, where the soil loss rate is approximately 5–10 times that of dry areas (Duck, 1996). Thus, it is significant to estimate rainfall erosivity to elucidate the microphysical characteristics of rainfall and rainfall–soil interactions. Benaud et al. (2020) collated empirical soil erosion observations from UK-based studies into a geodatabase. However, there is a limitation that this database does not cover the entirety of the UK, especially the limited records in northern Scotland. In our future work, we propose to compare the soil loss database with our estimated soil loss using WRF DSD based rainfall erosivity and a soil erosion model (such as RUSLE). We believe that not only can we better analyze the impact of rainfall and rainfall erosivity on the UK soil loss, but also help to better understand microphysical rainfall–soil interactions to support the rational formulation of soil and water conservation planning."

Benaud, P., Anderson, K., Evans, M., Farrow, L., Glendell, M., James, M. R., ... & Brazier, R. E. (2020). National-scale geodata describe widespread accelerated soil erosion. Geoderma, 371: 114378.

Ćurić, M., Janc, D., Vučković, V. and Kovačević, N. (2009). The impact of the choice of the entire drop size distribution function on Cumulonimbus characteristics. Meteorologische Zeitschrift 18(2): 207-222.

Duck, R. W. (1996). Regional variations of fluvial sediment yield in eastern Scotland. Erosion and Sediment Yield: Global and Regional Perspectives: Proceedings of an International Symposium Held at Exeter, UK, IAHS.

Kumar, P., Kishtawal, C. and Pal, P. (2017). Impact of ECMWF, NCEP, and NCMRWF global model analysis on the WRF model forecast over Indian Region. Theoretical and Applied Climatology 127(1-2): 143-151.

Liu, J., Bray, M. and Han, D. (2013). Exploring the effect of data assimilation by WR-F‐3DVar for numerical rainfall prediction with different types of storm events. Hydrological Processes 27(25): 3627-3640.

Pickering, B. S., Neely III, R. R., & Harrison, D. (2019). The Disdrometer Verification Network (DiVeN): a UK network of laser precipitation instruments. Atmospheric Measurement Techniques 12: 5845-5861.

Thompson, G. and Eidhammer, T. (2014). A study of aerosol impacts on clouds and precipitation development in a large winter cyclone. Journal of the Atmospheric Sciences 71(10): 3636-3658.

Yang, Q., Dai, Q., Han, D., Chen, Y. and Zhang, S. (2019). Sensitivity analysis of raindrop size distribution parameterizations in WRF rainfall simulation. Atmospheric Research 228: 1-13.

Point 5: What about ground truthing validation of your results in the whole UK or using previous studies with experimental and in-situ data?

Response 5: Rainfall erosivity are difficult to measure, because they refer to erosive potential of rainfall, not the amount of soil erosion that rainfall specifically causes. In RUSLE, soil loss can be estimated by multiplying the rainfall erosivity factor (R-factor) by five other factors: soil erodibility (K-factor), slope length (L-factor), slope steepness (S-factor), crop type and management (C-factor), and supporting conservation practices (P-factor). For ground verification, we believe that disdrometer is the most accurate measurement instrument currently for rainfall erosivity estimation. Results derived by disdrometers are sufficient as a reference to support this study. Moreover, DiVeN you pointed out in Point 2 may be a great data source for ground verification in our future in-depth work.

Point 6: How much your study can be compared or can support the very recently published research entitled "National-scale geodata describe widespread accelerated

soil erosion" https://doi.org/10.1016/j.geoderma.2020.114378. The latter publication can enrich the discussion part of the study.

Response 6: Thanks for your kind advice. As mentioned in the Point 5, rainfall erosivity and soil erosion caused by rainfall are completely different concepts. The publication you pointed out collected all readily available and empirically-derived soil erosion data from UK-based studies into a geodatabase. However, the database did not cover the entire UK completely. For instance, compared to England data, Scotland has very few soil erosion records in the database. Based on the analysis of existing records, authors found that there was a weak positive relationship between the total annual precipitation and soil erosion rates in some areas. We believe that putting the rainfall erosivity estimation based on WRF DSD into a soil erosion model (such as RUSLE) and estimating large-scale soil loss can enrich the UK soil loss database. In this way, not only can we better analyze the impact of rainfall and rainfall erosivity on UK soil loss, but also help to better understand microphysical rainfall–soil interactions to support the rational formulation of soil and water conservation planning.

The corresponding text has been added (see Point 4).

Point 7: Avoid using the abbreviation in the abstract and key points.

Response 7: Agreed and amended.

Point 8: Enrich the Figures and Tables captions, ensuring selfexplaining to the readers without referring to the main text and avoiding abbreviations.

Response 8: Agreed and amended.
* * *
[Figure]

**Fig. 1.** (new Figure 2 in manuscript). Minutes number per intensity class (x-axis) and ke class (y-axis) with five fitted ke–I curves at Chilbolton station (2004–2013), plotted on linear (left) and logarithmic

[Figure]

**Fig. 2.** (new Figure 4 in manuscript). Minutes number per intensity class (x-axis) and ke class (y-axis) with fitted ke–I curves at Bristol station (2015–2018), plotted on linear (left) and logarithmic (right)

---

## Author Response (AR2)

**Response to Referee #1 Comments**

This paper proposed a novel approach evaluating the soil erosion based on the rainfall dropsize distribution from WRF. Authors have addressed most of my comments. The only weak part is the discussion section. It would be completer and more meaningful if additional detailed comparisons with previous studies can be done to demonstrate the advantages of the proposed approach over the traditional approaches. Also it would be better to separate the discussion and conclusion section in my opinion. Other than this comment, the manuscript is good.

**Response:** The discussion and conclusion section were separated. The new conclusion section was rewritten as follows.

"This study presented a novel method for large-scale rainfall KE and erosivity estimation based on high-resolution, WRF-derived DSDs. Three microphysical parameterizations schemes (Morrison, WDM6, and Thompson aerosol-aware [TAA]) were designed to obtain raindrop size distributions, rainfall KE and rainfall erosivity for the entire of the UK covering the period of 2013–2017. With validation from the long-term observations of a disdrometer, the WRF-based rainfall erosivity exhibited an acceptable performance at Chilbolton station. Among the three WRF schemes, TAA exhibited the most superior performance and was recommended for future investigation. The results revealed that high rainfall erosivity occurred in the west coast area of the UK. Compared with the traditional empirical method, the proposed method can explain rainfall erosivity from a microphysical perspective and reflect more spatial variation because of changes in rainfall KE at the whole-country scale. Therefore, the development of a numerical weather prediction model offers an opportunity to better understand rainfall erosivity directly from its true definition. More importantly, because the WRF model is able to be driven by the global reanalysis data to obtain large-scale rainfall kinetic information, the proposed scheme can be easily applied to other regions, especially in ungauged areas.

Some problems remain with the proposed scheme, as discussed in section 5. Some of the problems, such as temporal downscaling of rainfall and point-to-area representative error by WRF, may introduce further uncertainty. This should be put in perspective of future work. It is expected that further exploration of research areas with different climatic and geographical characteristics would help us to establish a greater degree of accuracy on this matter."

In addition, the detailed comparisons with previous studies were added in the discussion section as follows:

"Compare to the previous large-scale rainfall erosivity studies based on empirical formula and spatial interpolation, this study presents a WRF-driven approach directly using the simulated rainfall microphysical variables. As demonstrated in the literatures, the relation between rainfall intensity and erosivity is not straightforward (Panagos et al., 2015a; Ballabio et al., 2017; Panagos et al., 2017). However, although all works show that rainfall erosivity decrease from west to east in UK, previous studies (Panagos et al., 2015a; Naipal et al., 2015) using traditional methods lead to an overestimation of rainfall erosivity, which may due to parameter $a$ in the universal $KE-I$ relationship is too high for the UK. Considering the five years (2013–2017) as a whole, the averaged $R_{W-Morrison}$, $R_{W-WDM6}$, and $R_{W-TAA}$-factor in each grid can be calculated. Nationally, the mean values of the three $R_W$-factors are 446.57, 640.92, and 416.35 MJ mm ha$^{-1}$ h$^{-1}$ y$^{-1}$, and their coefficients of variation (CV) are 0.56, 0.81, and 0.59, respectively. Compared with the outcomes (mean R-factor=746.6 MJ mm ha$^{-1}$ h$^{-1}$ y$^{-1}$, CV=0.81) of the Panagos et al. (2015a) using traditional methods, the R-factor of WDM6 scheme are quite similar, while other schemes have relatively low R-factors and low CVs."

**Response to Referee #2 Comments**

The discussion part of the paper can be further enriched. Better to separate the Discussion and Conclusion parts. The soil erosion yield is well known by its complexity and heterogeneity, especially at the field scale due to a large number of controlling factors and their interactions (such as surface roughness, land cover, soil properties etc.) during the erosive events. Benaud et al. 2020 (https://doi.org/10.1016/j.geoderma.2020.114378) is a very good example for that where the erosion rates between arable and grassland did not differ a lot considering different experiments conducted in different locations, times and for different purposes. However, under controlled conditions or theoretical framework (like the present study), the results look smoother, consistent and very promising. However, when they are confronted with the natural conditions (which are much complex), the conclusions change significantly. I still believe that the recent work of Benaud et al. 2020 (https://doi.org/10.1016/j.geoderma.2020.114378) can be used to validate the approach and/or (at least) enrich the discussion significantly. For instance, they clearly mentioned and showed how much the topsoil texture map (Fig. 3) could explain the spatial distribution and magnitude of soil erosion records of UK. Fig. 3 of Benaud et al. 2020 clearly also showed the correlation between the rainfall distribution (Rainfall distribution of UK) and soil erosion, which is very complementary to your statement. So, here I recommend the authors to emphasize the added value of this study to the soil erosion community by enriching the discussion part, significantly, considering the most relevant references.

**Response:** It is worth remarking that rainfall erosivity is not the soil eroding amount by rainfall, but a kind of erosion potential of the soil powered by rainfall kinetic energy. In terms of the influence of complex natural conditions such as vegetation, topography, and soil properties on soil erosion, they need to be considered in the soil erosion model. For example, the widely used RUSLE model takes into account these influences with six factors (rainfall erosivity factor R, soil erodibility factor K, soil length factor L, slope steepness factor S, cover management factor C, and supporting practices factor P). We suppose the comparison of soil erosion models is a significant research content but different from the aim of this work. In the future study, we will combine the WRF-derived rainfall erosivity results with the soil erosion model, and conduct an in-depth comparative analysis with Benaud et al. 2020 to investigate more physical mechanism of soil erosion.

The discussion and conclusion section were separated and additional detailed comparisons with previous studies were added in the discussion section (see Response to Referee #1).

[revised manuscript text omitted]

Compare to the previous large-scale rainfall erosivity studies based on empirical formula and spatial interpolation, this study presents a WRF-driven approach directly using the simulated rainfall microphysical variables. As demonstrated in the literatures, the relation between rainfall intensity and erosivity is not straightforward (Panagos et al., 2015a; Ballabio et al., 2017; Panagos et al., 2017). However, although all works show that rainfall erosivity decrease from west to east in UK, previous studies (Panagos et al., 2015a; Naipal et al., 2015) using traditional methods lead to an overestimation of rainfall erosivity, which may due to parameter $a$ in the universal $KE–I$

relationship is too high for the UK. Considering the five years (2013–2017) as a whole, the averaged $R_{W-Morrison}$, $R_{W-WDM6}$, and $R_{W-TAA}$-factor in each grid can be calculated. Nationally, the mean values of the three $R_W$-factors are 446.57, 640.92, and 416.35 MJ mm ha$^{-1}$ h$^{-1}$ y$^{-1}$, and their coefficients of variation (CV) are 0.56, 0.81, and 0.59, respectively. Compared with the outcomes (mean R-factor=746.6 MJ mm ha$^{-1}$ h$^{-1}$ y$^{-1}$, CV=0.81) of the Panagos et al. (2015a), the R-factor of WDM6 scheme are quite similar, while other schemes have relatively low R-factors and low

CVs.

[revised manuscript text omitted]

Naipal, V., Reick, C. H., Pongratz, J. and Van Oost, K. (2015). Improving the global applicability of the RUSLE model-adjustment of the topographical and rainfall erosivity factors. Geoscientific

Model Development 8: 2893-2913.

Nyssen, J., Vandenreyken, H., Poesen, J., Moeyersons, J., Deckers, J., Haile, M., Salles, C. and

Govers, G. (2005). Rainfall erosivity and variability in the Northern Ethiopian Highlands. Journal of Hydrology 311(1-4): 172-187.

O'Neill, D. (2007). The total external environmental costs and benefits of agriculture in the UK.

Environment Agency, UK.

Panagos, P., Ballabio, C., Borrelli, P., Meusburger, K., Klik, A., Rousseva, S., Tadić, M. P.,

Michaelides, S., Hrabalǩová, M. and Olsen, P. (2015a). Rainfall erosivity in Europe. Science of the Total Environment 511: 801-814.

Panagos, P., Borrelli, P., Meusburger, K., Yu, B., Klik, A., Lim, K. J., Yang, J. E., Ni, J., Miao, C.

and Chattopadhyay, N. (2017). Global rainfall erosivity assessment based on high-temporal resolution rainfall records. Scientific reports 7(1): 1-12.

[revised manuscript text omitted]